

# Diversity and distribution of the caddisfly genus *Atopsyche* Banks, 1905 in Ecuador, with the description of seven new species (Trichoptera: Hydrobiosidae)

Ernesto Rázuri-Gonzales[1,2], Ralph Holzenthal[1] and Blanca Ríos-Touma[3]

[1] Department of Entomology, University of Minnesota, Saint Paul, MN, United States of America
[2] Senckenberg Research Institute and Natural History Museum Frankfurt, Frankfurt am Main, Germany
[3] Facultad de Ingenierías y Ciencias Aplicadas. Ingeniería Ambiental. Grupo de Investigación en Biodiversidad, Medio Ambiente y Salud-BIOMAS, Universidad de las Americas, Quito, Ecuador

## ABSTRACT

*Atopsyche* is the largest Hydrobiosidae genus on the South American continent. The genus previously included 27 species in Ecuador. In this work, we describe and illustrate seven new species of *Atopsyche* from the Andes of Ecuador: *Atopsyche andina* **sp. nov.**, *Atopsyche azuayana* **sp. nov.**, *Atopsyche chocoandina* **sp. nov.**, *Atopsyche jocotoco* **sp. nov.**, *Atopsyche papallacta* **sp. nov.**, *Atopsyche piburja* **sp. nov.**, and *Atopsyche tapichalaca* **sp. nov.** Additionally, we provide distributional information for all Ecuadorian *Atopsyche*, including three new country records: *A. kingi*, *A. mayucapac*, and *A. neotropicalis*. With these additions, there are now 37 species of *Atopsyche* in Ecuador, or about 75% of species in the country, according to the CHAO 2 species estimator. Finally, we provide new and more detailed illustrations for *A. bolivari*, *A. bravoi*, and *A. davidsoni*.

Corresponding author
Blanca Ríos-Touma,
briostouma@gmail.com

## INTRODUCTION

Since 2011, we have been investigating the diversity and biology of the caddisflies of Ecuador (*Ríos-Touma et al., 2017*). To date, we have recorded 493 species from the country, based on about 50,000 curated and databased specimens from 250 collection events. Twenty-four new species and one new genus have been described so far in the families Anomalopsychidae, Polycentropodidae, Leptoceridae, and Philopotamidae (*Holzenthal & Rázuri-Gonzales, 2011*; *Holzenthal & Ríos-Touma, 2012*; *Camargos, Ríos-Touma & Holzenthal, 2017*; *Rázuri-Gonzales, Holzenthal & Ríos-Touma, 2017*; *Rázuri-Gonzales, Holzenthal & Ríos-Touma, 2018*; *Holzenthal, Blahnik & Ríos-Touma, 2018*; *Holzenthal, Blahnik & Ríos-Touma, 2022*). Here, we provide information on the diversity and distribution of 37 species of *Atopsyche*, family Hydrobiosidae, for the country, including the description of seven new species, three new country records, and the redescription of three others. We also assess the diversity

and elevational distribution of the species in Ecuador based on 884 specimens distributed among the 28 species we collected and provide estimates of predicted species richness.

The genus *Atopsyche* is endemic to the Western Hemisphere, where 146 extant and one fossil species occur, usually in mountainous regions, from the southwestern USA, Mexico, and Central America, the Greater Antilles, and most of South America, except the lowlands of Amazonia and the Patagonian region of Chile and Argentina, where several other endemic genera of hydrobiosids occur. Most species are regionally restricted if not highly endemic. Their aquatic larvae prefer cool water rivers and streams of moderate to high flow rates (*Gomes & Calor, 2016*). Hydrobiosids and the closely related family Rhyacophilidae do not build larval cases or retreats, unlike almost all other caddisflies. These "free-living" caddisflies crawl on submerged substrates in flowing waters, feeding as predators on other aquatic macroinvertebrates. The foreleg tibia and tarsus in *Atopsyche* and other hydrobiosids are modified into a chelate pinching structure that aids in capturing prey (*Springer, 2010*). Final instar larvae build a dome-shaped pupal enclosure of small mineral fragments and spin a silken pupal cocoon inside which they pupate. Adults emerge and become aerial. Little is known about adult behavior or biology, except that they are attracted to ultraviolet lights during evening hours.

Traditionally, the genus *Atopsyche* has been further divided into three subgenera: *Atopsyche, Atopsaura Ross, 1953*, and *Dolochorema Banks, 1913*. *Schmid (1989)* later established the *A. bicolorata* species group for three species that resembled members of *Dolochorema,* but that could not be included in any of these subgenera. Finally, *Schmid (1989)* mentioned the presence of two "isolated" species. These subgenera were established based on features of the inferior appendages and, to a lesser extent, the phallic apparatus, with the 30 species known at the time (*Ross, 1953*; *Schmid, 1989*). However, several characteristics of the phallic apparatus, for example the absence or presence of the ventrolateral branches and the basodorsal processes (*i.e.,* Schmid's dorsal rod) of the phallic apparatus, contradict these groups. *Schmid (1989)* in his revision of the Hydrobiosidae placed the 45 species he described in existing subgenera. However, he and later *Blahnik & Gottschalk (1997)* recognized the need for an updated subgeneric classification of this genus. Herein, we describe and illustrate seven new *Atopsyche* species from Ecuador without placing them into subgenera. We additionally provide distributional information for all Ecuadorian *Atopsyche,* including three new country records: *A. kingi*, *A. mayucapac*, and *A. neotropicalis*. Finally, we provide new and more detailed illustrations for *A. bolivari, A. bravoi*, and *A. davidsoni*.

## MATERIALS AND METHODS

We collected adult specimens using Leciel UV black 10W LED lights positioned proximate to stream environments. These lights were tethered to an Inui USB power pack (22.5 W, 20,000 mAh) and hung in front of a white bed sheet. Additional lights were placed upon nearby white trays containing ethanol and also connected to a USB power pack. The lights were on for 2–3 h at dusk. Specimens collected on the sheet were killed using an ammonium carbonate kill jar and then pinned using entomological pins, while the specimens collected

in the trays were fixed in ethanol 80%. Collecting permits "*BIODIVERSIDAD DE ORGANISMOS DE AGUA DULCE DEL ECUADOR*" No. MAAE-DBI-CM-2021-0161 and 003-14-1CFAU-FLO-DNB/MA authorized us to collect the specimens.

## Specimen preparation and observation

Adult specimens were examined and prepared using conventional techniques for alcohol-preserved and dry specimens (*Blahnik & Holzenthal, 2004*; *Blahnik, Holzenthal & Prather, 2007*). Forewing length was measured from the base to the apex and reported as a range alongside the number of specimens measured. Male genitalia were macerated in 85% lactic acid at 125 °C for 20 min to dissolve internal soft tissues. Females of only six of the 154 species in the genus have been described. In our study, females collected at the same time and place as males, and that matched males in size and color pattern, are included as paratypes. However, these should be considered tentative associations until female morphological characters have been studied comparatively across the genus or DNA sequence data (*e.g.*, COI barcodes) have been assessed to establish associations. Similarly, immature stages have been described for only 11 species. Among the Ecuadorian species, the larval and pupal stages of only *A. callosa* are known (*Rueda-Martín, 2006*).

## Illustrations and descriptions

The specimens were illustrated using an Olympus BX41 compound microscope with a U-DA drawing tube at 200× and 400× magnification. Additionally, an Olympus SZX12 stereo zoom microscope at 90–144× magnification was used to verify details in the illustrations. The pencil sketches of the genitalic structures were then scanned and imported into Adobe Illustrator Creative Cloud version 29.0 to serve as a template for creating vector illustrations. Illustrations of new and previously described species are presented in Figs. 1–10.

The distribution map was prepared in QGIS 3.36.0 Maidenhead (*QGIS Development Team, 2024*) using vector and raster data from *Natural Earth (2018)* and CIAT-CSI SRTM (*Jarvis et al., 2008*), respectively. The resulting distributions of the species described are presented in Fig. 11.

## Morphological terminology

The morphological terminology follows *Schmid (1989)*, except we prefer the term "phallic spine" rather than "aedeagus" since the later is used for the entire phallus by some authors or just the distal part by others (*De la Torre-Bueno, 1989*). For simplicity, paired structures are described in the singular. Each specimen was affixed with a barcode label bearing a unique alphanumeric sequence beginning with the prefix UMSP, serving as an exclusive identifier for specimen data uploaded to the University of Minnesota Insect Collection (UMSP) *Specify* database. These data are also available in GBIF.

## New species names

The electronic version of this article in portable document form will represent a published work according to the International Commission on Zoological Nomenclature (ICZN). Hence, the new names in the electronic version are effectively published under that Code from the electronic edition alone. This published work and the nomenclatural acts it
contains have been registered in ZooBank, the online registration system for the ICZN. The ZooBank Life Science Identifiers (LSIDS) can be resolved, and the associated information can be viewed through any standard web browser by appending the LSID to the prefix http://zoobank.org/. The LSID for this publication is urn:lsid:zoobank.org:pub:53F3F8F1-8F73-4FDD-B5FC-5F929AE66411. The online version of this work is archived and available from the following digital repositories: PeerJ, PubMed Central, and CLOCKSS.

## Diversity and distribution analysis

We analyzed the geographical distribution of Ecuadorian *Atopsyche* based on the specimens collected and recorded in the UMSP *Specify* database. Using species occurrences, we calculated the CHAO 2 species estimator in its biased corrected form (*Gotelli & Colwell, 2011*) to assess the potential species richness of this genus in the country. We only included our collections because each locality had the same sampling effort and included complete georeferenced data unavailable from old literature records.

# RESULTS

## Species descriptions

*Atopsyche andina*, Rázuri-Gonzales, Holzenthal & Ríos-Touma sp. nov.
LSID urn:lsid:zoobank.org:act:E2AC6B4D-F727-43E2-A458-3D1C2AB58B29
Fig. 1

## Diagnosis

This new species (Fig. 1) is most similar to *A. davidsoni Sykora, 1991*, which he placed in the *bicolorata* group of *Schmid (1989)* and was said to be close to *A. chirimachaya Harper & Turcotte, 1985*. However, Schmid only included *A. bicolorata*, *A. unicolorata*, and *A. yupanqui* in the *bicolorata* group. *Atopsyche chirimachaya* along with *A. cajas* were listed by Schmid as "espèces isolées". Despite the possible erroneous placement of *A. davidsoni*, *A. andina* is most similar to *A. davidsoni* based on our examination and comparison with the holotype of Sykora's species, here re-illustrated (Fig. 3). The new species shares with *A. davidsoni* similarly shaped parapods and a pair of mesoventral spines on the phallotheca. The inferior appendages of both species have the first segment short and quadrate and the second segment short, triangular, and slightly curved mesad. In *A. andina,* there is an elongate digitate apicoventral process which is only slightly developed in *A. davidsoni.* Mesoventrally, *A. andina* bears a well-developed spine-like process that is less developed in *A. davidsoni*. Lastly, the spines on the mesal face of the inferior appendage are stronger and differently arranged in *A. andina* than in *A. davidsoni.*

## Material examined
### Holotype

ECUADOR ● ♂; Bolivar Province, Río Puracachi, 2 km (air) NW Salidas de Guaranda; 1.39328°S, 79.03575°W; 3615 m a.s.l.; 19 Mar. 2022; B. Ríos, R. Holzenthal, S. Pauls, R. Thomson and X. Amigo leg.; UMSP (pinned) [UMSP000503585].

## Paratypes

ECUADOR ● 1♂1♀; same data as the holotype; MECN (pinned) ● 14♂♂2♀♀; same data as the holotype; UMSP (pinned) ● 1♂; Pichincha Province, Reserva Paluguillo, Quebrada Saltana; 0.31644°S, 78.2032°W; 3850 m a.s.l.; 17 Dec. 2009; F. González and B. Ríos-Touma leg.; SMF (in alcohol) ● 1♂; same data as the preceding; 30 Mar. 2012; L. Pita and B.Ríos-Touma leg.; UMSP (pinned) ● 1♂1♀; same data as the preceding; 15–16 Oct. 2011; R. Holzenthal, L. Pita and B. Ríos-Touma leg.; UMSP (pinned) ● 1♂; same data as the preceding; 14–18 Apr. 2011; B. Ríos-Touma leg.; UMSP (in alcohol) ● 3♂♂; same data as the preceding; 22 Jul. 2009; González and B. Ríos-Touma leg.; UMSP (in alcohol) ● 1♂; same data as the preceding; 19 Nov. 2009; B. Ríos-Touma leg.; UMSP (in alcohol) ● 19 ♂♂; Napo Province, Parque Nacional Cayambe-Coca, Río Papallacta, above Termas Papallacta; 0.35364°S, 78.15117°W; 3395 m a.s.l.; 17 Nov. 2023; B. Ríos, R. Holzenthal, P. Frandsen and X. Amigo leg.; UMSP (in alcohol).

## Description

**Adult.** Forewing length: male (11–16 mm, $n = 10$), female (12.5–15.5 mm, $n = 4$). Body and wings dark brown. Forewing with scattered brown setae and longer, erect straw-colored setae along major longitudinal veins; with patch of longer, erect cream-colored setae at base of anal veins and adjacent much smaller patch of erect black setae apical to it; patch of cream-colored setae at apex of Cu2; narrow irregular row of darker setae bordering posterior edge between cream-colored patches; very small patches of cream-colored setae at apices of major longitudinal veins at apex of wing from R1 through M4. Sterna III–IV without glands; sternum V with a pair of long, membranous glands; processes on sternum VI and VII short.

**Male genitalia.** Segment IX, in lateral view, quadrangular, almost as high as long, with setae on posteroventral surface. Parapod, in lateral view, elongate, same width throughout its length, slightly bent ventrad, apex rounded, apicoventrally with digitate lobe, dorsal surface with a few setae subapically; in dorsal view, elongate, lateral margin slightly concave basally, mesal margin slightly concave subapically, setae on dorsal surface on apical half, apex rounded. Filipod digitate, longer than parapods, setose. Preanal appendage short, rounded, setose. First segment of inferior appendage, in lateral view, rectangular, ventral margin slightly sinuous, dorsal margin slightly convex subapically, posteroventral corner produced into short, digitate process, with setae on ventral margin and lateral surface subapically; in ventral view, mitten-shaped, setose, lateral margin straight, mesal margin with an apical digitate projection (slightly narrower and arising subapically in paratype), a subtriangular projection mid-length with denticules apically (broader and without denticules in paratype), and a quadrate projection with irregular margins (narrower in paratype) (these projections vary within each side of a specimen and among specimens); second segment of inferior appendage, in lateral view, triangular, with a few setae on dorsal and ventral margins, dorsal and ventral margins slightly sinuous, apex acute; second segment of inferior appendage, in ventral view, digitate, curved (this segment is slightly bent in one of the paratypes), apex acute. Proctiger, in lateral view, narrow basally, wider apically, with a long carina laterodorsally, ventral and posterior margins

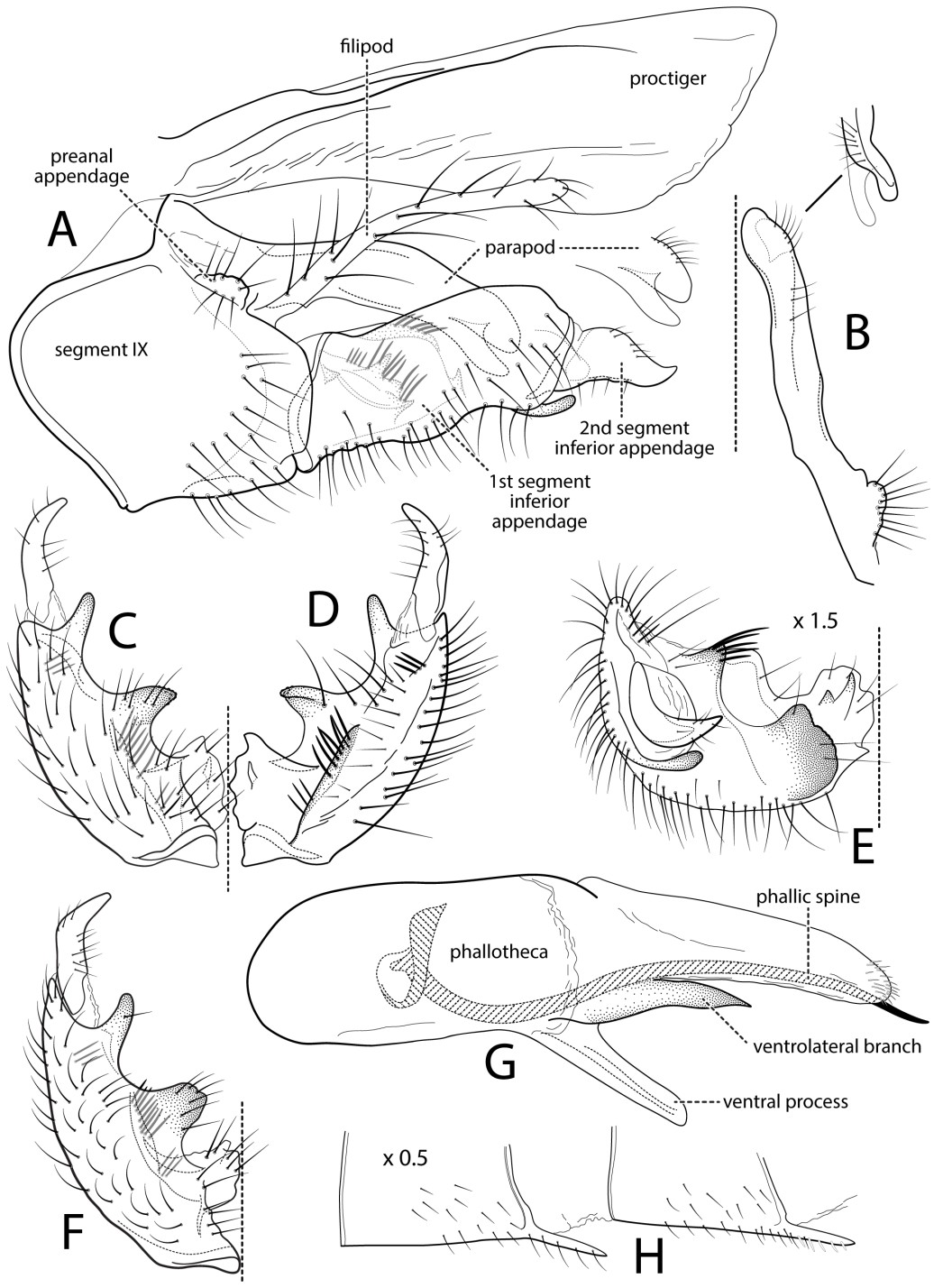

**Figure 1** *Atopsyche andina.* Male genitalia of *Atopsyche andina*, new species. (A) Segments IX and X, lateral (inset, apex of parapod, exposed). (B) Left parapod and preanal appendage, dorsal (inset, apex caudal). (C) Inferior appendage, ventral. (D) Inferior appendage, dorsal. (E) Inferior appendage, caudal (1.5×). (F) Inferior appendage, ventral (paratype). (G) Phallus, lateral. (H) Segments VI and VII, sternal processes (0.5×).

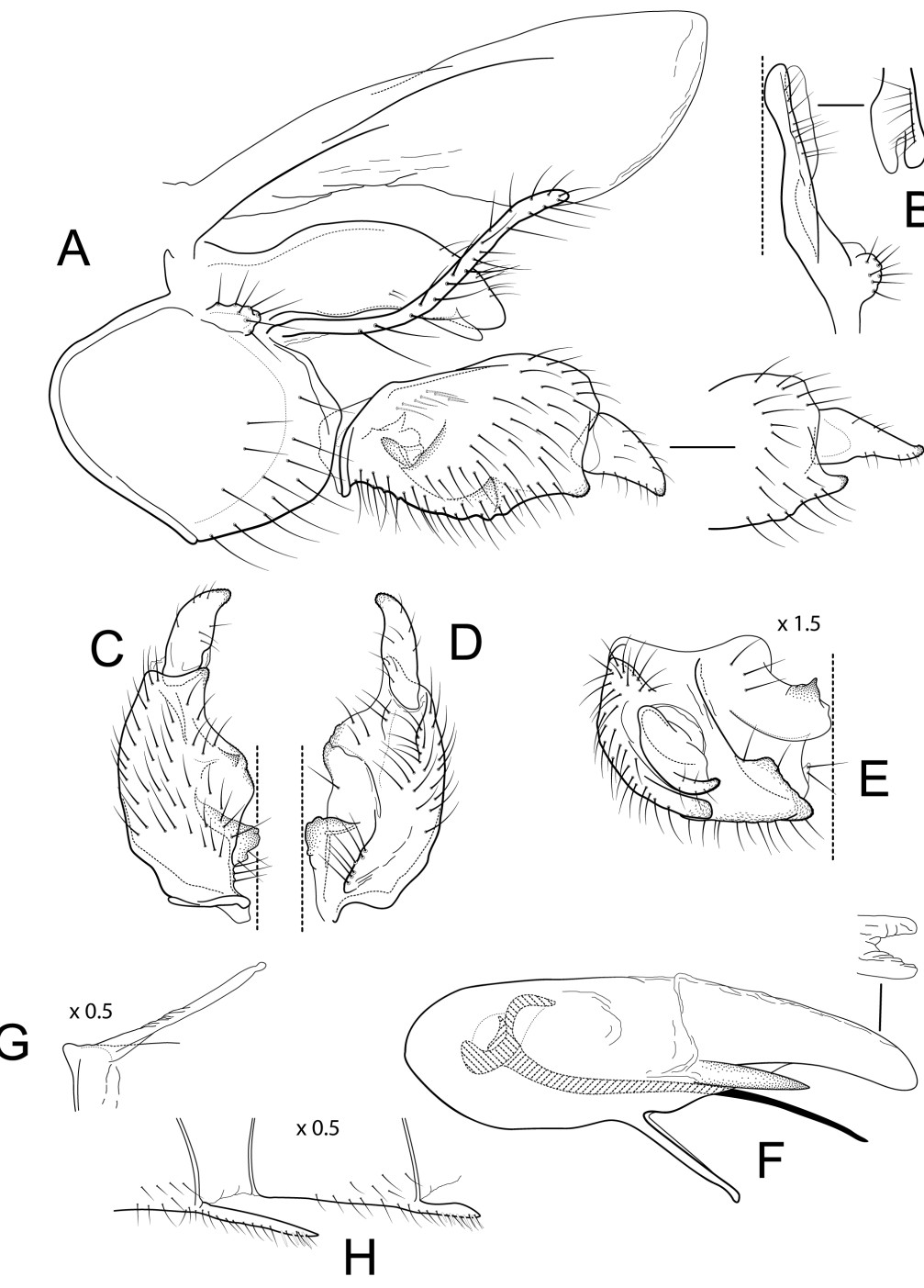

**Figure 2** *Atopsyche davidsoni.* Male genitalia of *Atopsyche davidsoni Sykora, 1991*. (A) Segments IX and X, lateral (inset, apex of inferior appendage, slightly rotated). (B) Left parapod and preanal appendage, dorsal (inset, apex caudal). (C) Inferior appendage, ventral. (D) Inferior appendage, dorsal. (E) Inferior appendage, caudal (1.5×). (F) Phallic apparatus, lateral. (G) Sternum V gland, lateral (0.5×). (H) Segments VI and VII, sternal processes (0.5×).

slightly membranous, without setae, apex truncate. Phallic apparatus complex; phallotheca broadly rounded basally, phallic apodeme indiscernible; with ventral process articulating with inferior appendages, narrow, same width throughout its length; ventrolateral branches of phallotheca present, triangular, approximately 0.55 times as long as posterior half of phallotheca, acute apically; dorsal process of phallotheca absent; posterior section of phallotheca, in lateral view, broad basally, tapering towards apex, directed posterad, apex covered with short setae, apex rounded; posterior section of the phallotheca, in dorsal view, with a shallow notch mesally; phallic spine elongate, stout, spine-like structure, with a slight convex curvature near the base, then slightly sinuous; in dorsal view, apex acute.

## Distribution
Ecuador: Bolivar, Napo, and Pichincha Provinces.

## Etymology
*Atopsyche andina* is named after the Andes mountains, where this species inhabits.

### *Atopsyche davidsoni* Sykora, 1991
Fig. 2
*Atopsyche davidsoni* (*unplaced*) Sykora, 1991:246 [Type locality: Ecuador, Prov. Bolivar, 16 km NNE Guaranda; CMNH; ♂].

## Material examined
### *Holotype*
ECUADOR ● ♂; Bolivar Province, 16 km NNE Guaranda; 3420 m a.s.l., 16 Oct. 1987; R. Davidson, J. Rawlins and C. Young leg.; CMNH.

### *Additional material*
ECUADOR ● 1♂1♀; Napo Province, Parque Nacional Cayambe-Coca, Río Papallacta, above Termas Papallacta; 0.35364°S, 78.15117°W; 3387 m a.s.l.; 17 Nov. 2023; B. Ríos, R. Holzenthal, P. Frandsen, X. Amigo leg.; UMSP (pinned) ● 1♂; same data as preceding; UMSP (in alcohol).

## Distribution
Ecuador: Bolivar Province.

### *Atopsyche azuayana* Rázuri-Gonzales, Holzenthal & Ríos-Touma, sp. nov.
LSID urn:lsid:zoobank.org:act:44110D77-4353-4664-8C73-E7207006DFD1
Fig. 4

## Diagnosis
The male genitalia of *Atopsyche azuayana* (Fig. 4) is rather simple, lacking the spines and processes seen on the parapods and inferior appendages of many species of *Atopsyche*. As

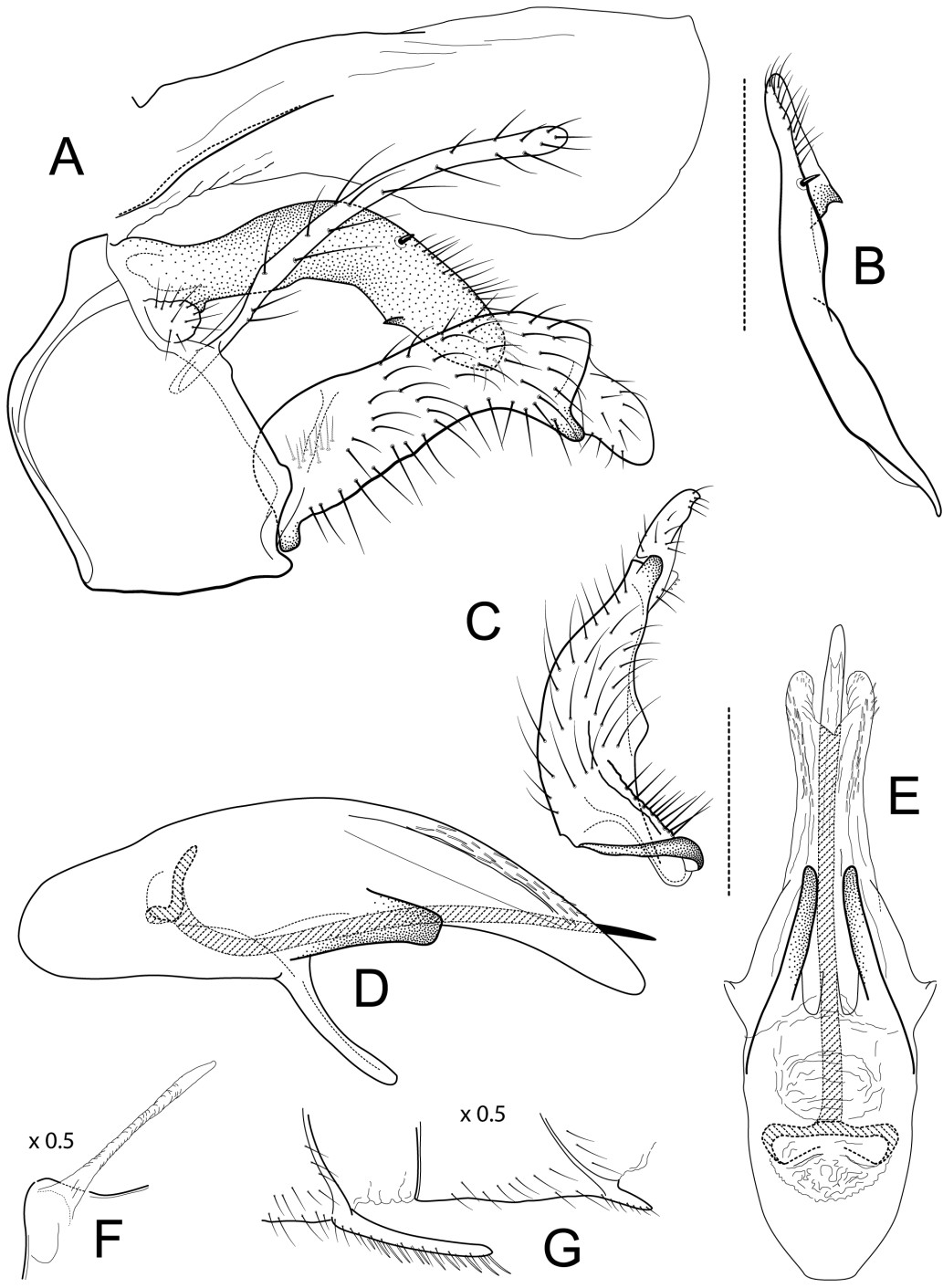

**Figure 3 _Atopsyche azuayana._** Male genitalia of _Atopsyche azuayana_, new species. (A) Segments IX and X, lateral (B) Left parapod dorsal. (C) Inferior appendage, ventral. (D) Phallic apparatus, lateral. (E) Phallic apparatus, dorsal. (F) Sternum V gland, lateral (0.5×). (G) Segments VI and VII, sternal processes (0.5×).

such, it is difficult to place it near any of the described species. Diagnostic features include the general shapes of the parapods and the inferior appendages with their convex and concave curvatures reminiscent of a boomerang.

## Material examined

### Holotype

ECUADOR ● ♂; Azuay Province, Río Angas, between Angas & Soldados; 2.88741°S, 79.31654°W; 3,645 m a.s.l.; 24 Mar. 2022; B. Ríos, R. Holzenthal, S. Pauls, P. Frandsen, R. Thomson and X. Amigo leg.; UMSP (pinned) [UMSP000503539].

### Paratypes

ECUADOR ● 1♀; same data as the holotype; UMSP (pinned) ● 1♂1♀; same data as the holotype; MECN (pinned).

## Description

**Adult.** Forewing length: male (12 mm, $n = 2$), female (13–13.5 mm, $n = 2$). Body and wings dark brown. Forewing with scattered brown setae and longer, erect brown- and straw-colored setae along major longitudinal veins; with small patch of erect, black setae where A2 joins A1; very small patch of cream-colored setae at apex of Cu2; very small patches of cream-colored setae at apices of major longitudinal veins at apex of wing from R1 through Cu1a. Sterna III–IV without glands; sternum V with a pair of long, membranous glands; process on sternum VI long, process on sternum VII very short.

**Male genitalia.** Segment IX, in lateral view, quadrangular, higher than long, dorsal margin short, without setae. Parapod, in lateral view, elongate, same width throughout its length, concave mid-length, slightly bent ventrad, apex rounded, with single peg-like setae dorsally, short setae dorsally on apical half, and a small spine-like projection ventrally on apicl fourth; in dorsal view, elongate, lateral margin slightly concave mid-length, then with a slight bump, and a sharp lateral projection, mesal margin very slightly sinuous, with setae and a single spine-like setae on apical fourth, apex narrowly rounded. Filipod digitate, longer than parapods, setose. Preanal appendage short, rounded, setose. First segment of inferior appendage, in lateral view, quadrate, ventral margin concave mid-length, dorsal margin slightly convex subapically, posteroventral corner produced into short, digitate process, with setae on ventral margin and lateral surface; in ventral view, sickle-shaped, basal margin slightly folded over mesal margin, setose, lateral margin convex, mesal margin with a small, subtriangular projection mid-length and a setose ridge basally, posterior margin rounded; second segment of inferior appendage, in lateral view, subtriangular, setose, dorsal and ventral margins straight, apex rounded; second segment of inferior appendage, in ventral view, digitate, straight, mesal margin with a small bump bearing denticules basally, apex narrowly rounded. Proctiger, in lateral view, narrow basally, wider apically, with a short carina laterally at the base, slightly membranous, without setae, apex slightly rounded. Phallic apparatus complex; phallotheca broadly rounded basally, phallic apodeme indiscernible; with ventral process articulating with inferior appendages, narrow, same width throughout its length; ventrolateral branches of phallotheca present, quadrate, approximately 0.4 times as long as posterior half of phallotheca, truncate apically; dorsal

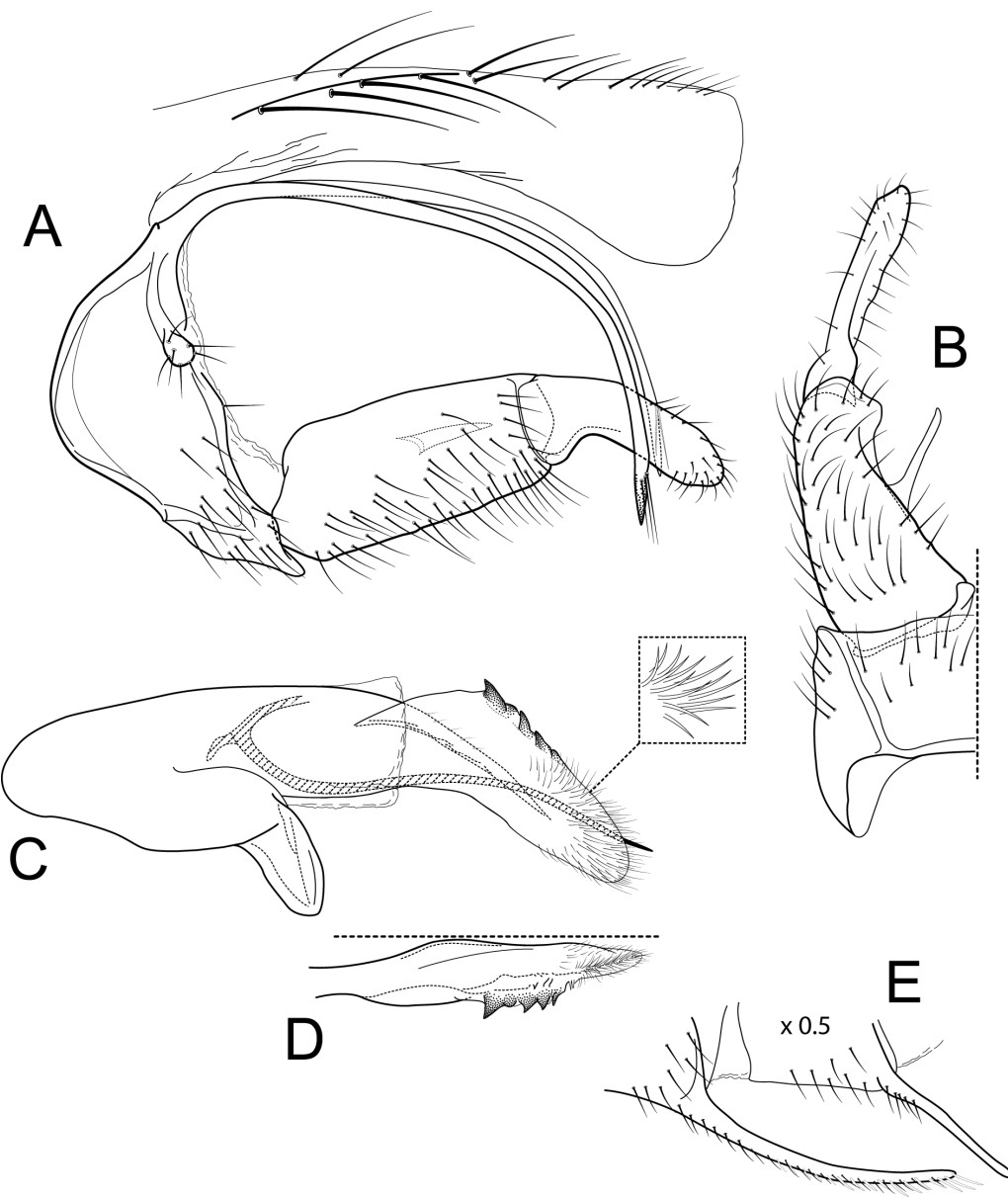

**Figure 4** *Atopsyche chocoandina.* Male genitalia of *Atopsyche chocoandina*, new species. (A) Segments IX and X, lateral. (B) Inferior appendage, ventral. (C) Phallic apparatus, lateral (inset, apical setae, enlarged). (D) Phallic apparatus, left side of phallotheca, dorsal. (E) Segments VI and VII, sternal processes (0.5×).

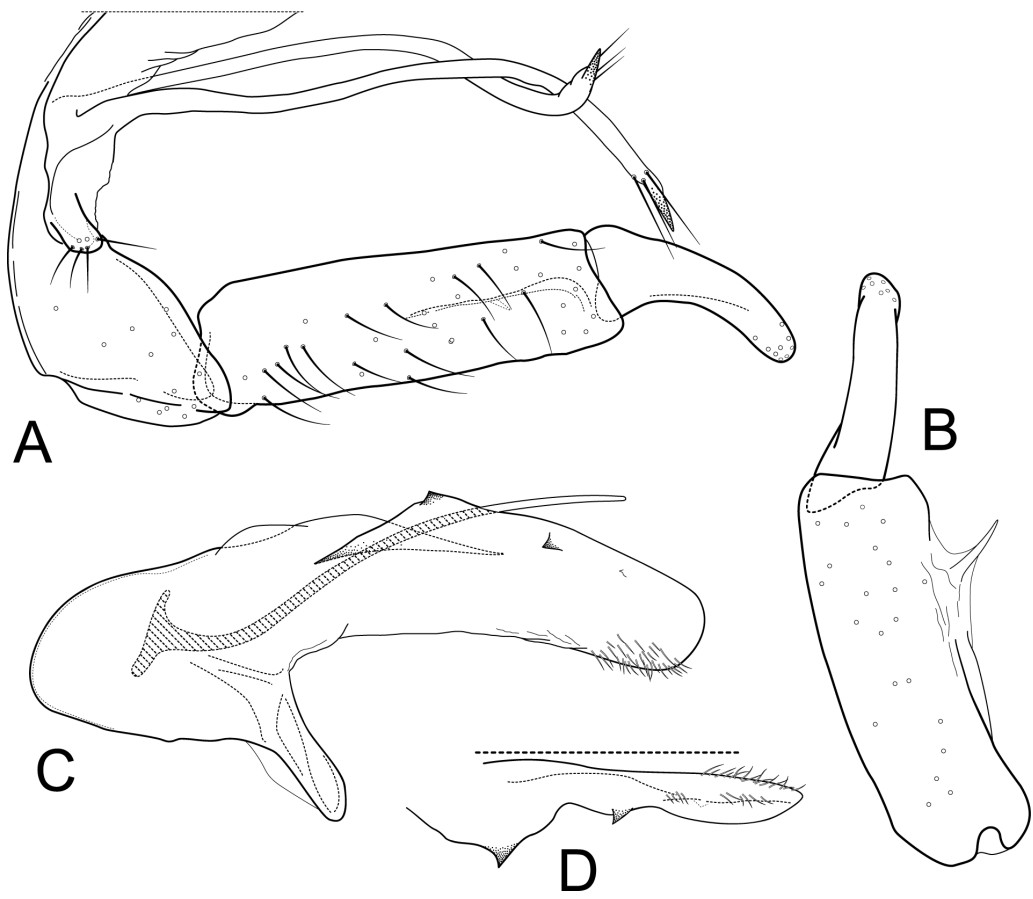

**Figure 5** *Atopsyche bolivari.* Male genitalia of *Atopsyche bolivari* Banks, 1924. (A) Segments IX and X, lateral. (B) Inferior appendage, ventral. (C) Phallic apparatus, lateral. (D) Phallic apparatus, left side of phallotheca, dorsal.

process of phallotheca absent; posterior section of phallotheca, in lateral view, broad basally, tapering towards apex, directed posteroventrad, dorsal surface covered with spine-like setae, apex narrowly rounded; posterior section of phallotheca, in dorsal view, with a shallow notch mesally; phallic spine elongate, stout, with a slight convex curvature near the base, then slightly sinuous; in dorsal view, apex acute.

## Distribution
Ecuador: Azuay Province.

## Etymology
*Atopsyche azuayana* is named after the Azuay province, where this species inhabits.

**Atopsyche chocoandina** Rázuri-Gonzales, Holzenthal & Ríos-Touma, sp. nov.
LSID urn:lsid:zoobank.org:act:F524722C-1CFD-4EFC-A37B-A277F038C3D1
Figs. 2, 5

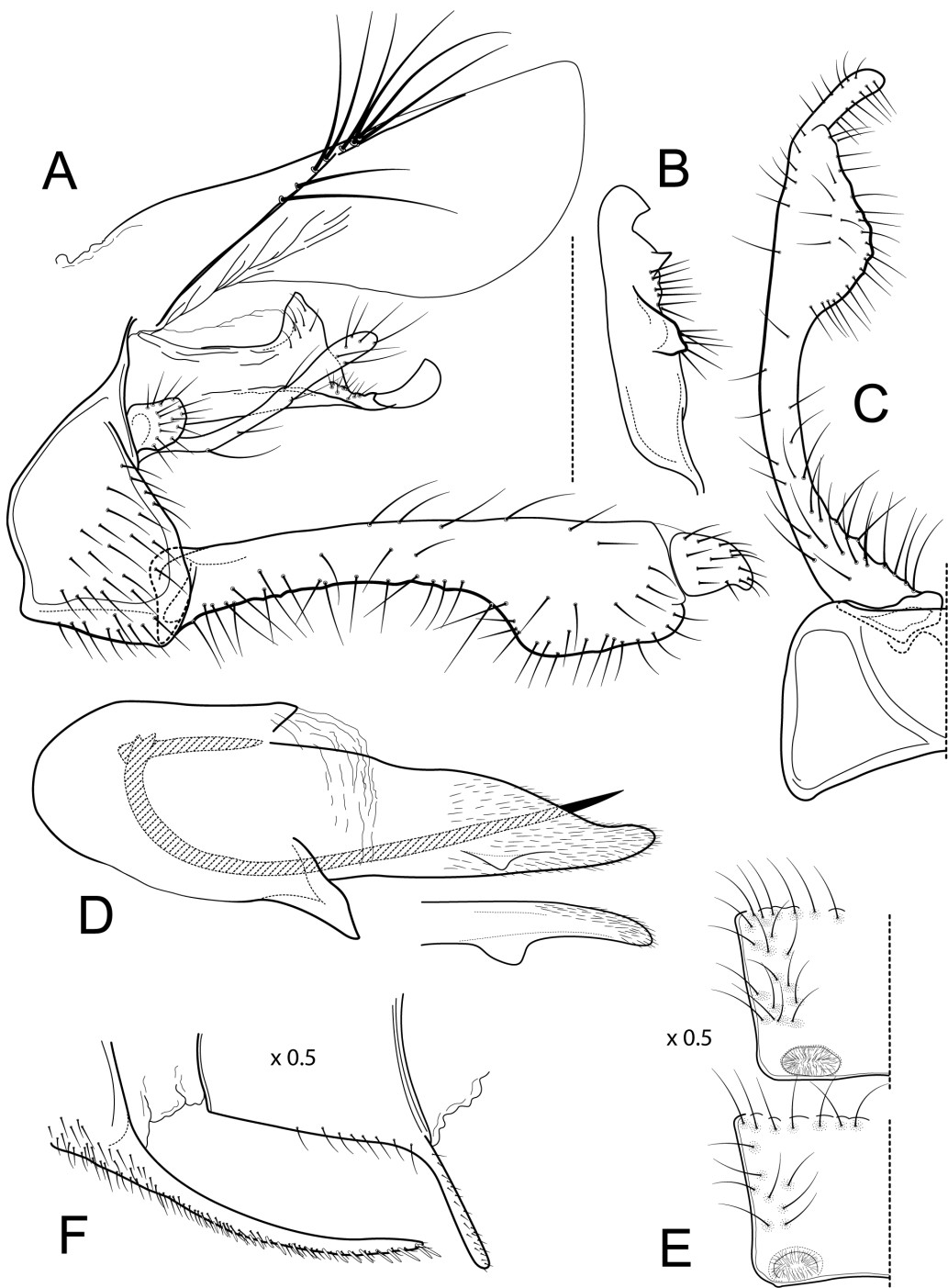

**Figure 6 *Atopsyche jocotoco*.** Male genitalia of *Atopsyche jocotoco*, new species. (A) Segments IX and X, lateral. (B) Left parapod, dorsal. (C) Inferior appendage and sternum IX, ventral. (D) Phallic apparatus, lateral (inset, left side of phallotheca, dorsal). (E) Terga III and IV glands, dorsal (0.5×). (F) Segments VI and VII, sternal processes (0.5×).

## Diagnosis

This new species (Fig. 5) is similar to *A. bolivari* (Fig. 6) described by *Banks (1924)* from Colombia and redescribed by *Ross & King (1952)* from the holotype. We also borrowed the holotype, here re-illustrated, to compare with *A. chocoandina*. Both species have narrow, very elongate parapods, but in *A. chocoandina* they are curved ventrad and are of equal length. In *A. bolivari*, they project more caudad and are less curved. The main difference is in the length of the first segment of the interior appendage, which is more elongate, about twice as long as the apical segment, in *A. bolivari*. In the new species, the basal segment is shorter and about equal to the length of the apical segment. Finally, there is a row of about 4–5 spines on the dorsal edge of the phallotheca of *A. chocoandina*, but only two such spines in *A. bolivari*.

## Material examined

### Holotype

ECUADOR ● ♂; Pichincha Province, Reserva el Cedral, small stream; 0.01146°S, 78.56945°W; 2,170 m a.s.l.; 12 Aug. 2017, A. Medina leg.; UMSP (in alcohol) [UMSP000278210].

### Paratypes

ECUADOR ● 3♀♀; same data as the holotype; UMSP (in alcohol) ● 1♂; same as the preceding; 14 Aug. 2017; UMSP (in alcohol) ● 2♂♂1♀; Bellavista Cloud Forest Reserve and Lodge, small stream; 0.01212°S, 78.68958°W; 2614 m a.s.l.; 14 Jul. 2017; A. Tapia leg.; MECN (in alcohol) ● 1♂; Bellavista Cloud Forest Reserve and Lodge, small stream on Trail F; 0.01629°S, 78.68241°W; 2,250 m a.s.l.; 23 Aug. 2020; B. Ríos-Touma and X. Amigo; SMF (in alcohol).

## Description

**Adult.** Forewing length male: 7.5 mm ($n = 4$), female: 8–10 mm ($n = 4$). Body and wings light brown. Wings denuded (specimens in ethanol). Sterna III–IV without glands; sternum V with a pair of tiny protuberances; process on sternum VI long and curved, process on sternum VII shorter than the process on sternum VI and straight.

   **Male genitalia.** Segment IX, in lateral view, quadrangular, much higher than long, dorsal margin very short, with setae on posteroventral surface. Parapod, in lateral view, elongate, very narrow throughout its length, strongly bent ventrad, apex acute, with a few short setae subapically; in dorsal view, very elongate and narrow, downturned, with setae subapically. Filipod absent. Preanal appendage short, rounded, setose. First segment of inferior appendage, in lateral view, rectangular, ventral and dorsal margins straight, posteroventral corner not produced, with setae on ventral margin and lateral surface; in ventral view, quadrate, setose, lateral margin very slightly convex, mesal margin very slightly concave with a slender projection mid-length, as long as half the length of the first segment of the inferior appendage, posterior margin truncate; second segment of inferior appendage, in lateral view, digitate, with a few setae apically, dorsal margin slightly convex, ventral margin concave, slightly inflated basally, apex rounded; second segment of inferior appendage, in ventral view, digitate, straight, apex rounded. Proctiger, in lateral view,

narrow basally, slightly wider apically, with a short carina laterodorsally at the base, slightly membranous, very long setae along basal carina and shorter setae along dorsal margin, apex truncate. Phallotheca broadly rounded basally, phallic apodeme indiscernible; with ventral process articulating with inferior appendages, thumb-shaped; ventrolateral branches of phallotheca absent; dorsal process of phallotheca absent; posterior section of phallotheca, in lateral view, broad throughout its length, directed posteroventrad, dorsal surface with spine-like sclerotizations mid-length, apical half covered with long, spine-like setae, apex rounded; posterior section of phallotheca, in dorsal view, with a deep notch mesally; phallic spine elongate, stout, with a slight convex curvature near base, then slightly sinuous; in dorsal view, apex acute.

## Distribution
Ecuador: Pichincha Province.

## Etymology
*Atopsyche chocoandina* is named after the Choco Andino Biosphere Reserve, where this species occurs, in honor of its biodiversity and the people who protect it.

### *Atopsyche bolivari* *Banks, 1924*
Fig. 6

*Atopsyche bolivari* (*Atopsyche*) *Banks, 1924*:443 [Type locality: Colombia, Dpto. Tolima, Monte Socorro, Tochecito, Quindini; MCZ; ♂]. —*Ross & King, 1952*:195 [♂]. —*Flint Jr, 1967*:2 [♂ lectotype]. —*Muñoz Quesada, 2000*:275 [checklist].

## Material examined
### *Lectotype*
COLOMBIA ● 1♂; Tolima Department, Monte Socorro, Tochecito, Quindini; Eduard Fassl leg.; MCZ [MCZ-ENT14839].

## Distribution
Colombia: Tolima Department.

### *Atopsyche jocotoco* Holzenthal, Rázuri-Gonzales & Ríos-Touma, sp. nov.
LSID urn:lsid:zoobank.org:act:9E54E406-AA5B-42F9-A304-D93D6AB815A0
Figs. 2, 7

## Diagnosis
This distinctive species (Fig. 7) is similar to *A. allani* *Holzenthal & Cressa, 2002*, as both have long first inferior appendage segments and very short second ones. They share a slight similarity in the structure of the parapod, but *A. allani* is more pronounced and spinose than *A. jocotoco*. *Atopsyche allani* has a dorsal spine-like process on the phallotheca, which is absent in *A. jocotoco*. The inferior appendage in lateral view is distinctly axe-shaped, a feature not seen in other species in the genus.

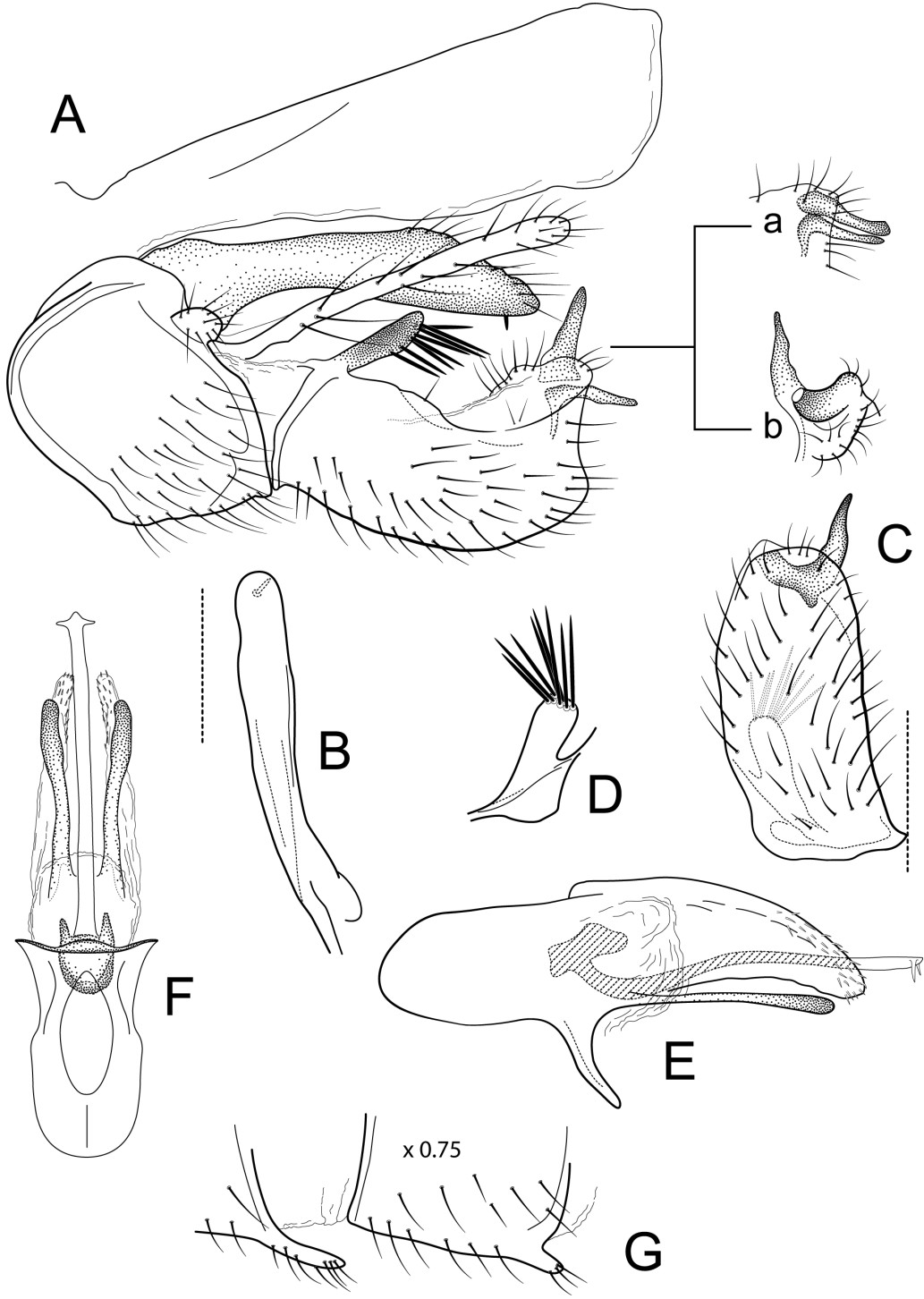

**Figure 7** *Atopsyche papallacta.* Male genitalia of *Atopsyche papallacta*, new species. (A) Segments IX and X, lateral (inset, processes at the apex on inferior appendage, a-lateral, appressed, b-caudal, apart). (B) Left parapod, dorsal. (C) Inferior appendage, ventral. (D) Inferior appendage, basodorsal process, dorsal. (E) Phallic apparatus, lateral. (F) Phallic apparatus, ventral. (G) Segments VI and VII, sternal processes (0.75×).

## Material examined

### Holotype
ECUADOR ● ♂; Zamora Chinchipe Province, Tributary to Quebrada Tapichalaca on Highway E682, N of Valladolid; 4.49422°S, 79.12828°W; 2,435 m a.s.l., 11 Feb. 2023; B. Rios, X. Amigo, J. Huisman leg.; UMSP (in alcohol) [UMSP000551054].

### Paratypes
ECUADOR ● 1♂; same data as the holotype; MECN (in alcohol).

## Description

**Adult.** Forewing length male: 7.5 mm ($n = 2$). Body light brown, wings dark brown. Wings denuded (specimens in ethanol). Sterga III–IV with oval glands, lined internally with spines; sternum V with a pair of tiny protuberances; process on sternum VI long and curved, process on sternum VII shorter than the process on sternum VI and straight.

**Male genitalia.** Segment IX, in lateral view, quadrangular, much higher than long, dorsal margin obliterated, with setae on posteroventral surface. Parapod, in lateral view, short, apical half narrower than basal half, with a triangular lobe dorsally and short lobe ventrally, directed posterad, apex narrowly rounded, with short setae on triangular lobe and ventral margin; in dorsal view, short, lateral margin with three spine-like projections, mesal margin straight, with setae subapically on lateral margin, apex rounded. Filipod digitate, slightly shorter or as long as parapods, setose. Preanal appendage short, rounded, setose. First segment of inferior appendage, in lateral view, apical third twice as high as basal section, ventral and dorsal margins straight, posteroventral corner not produced, with setae on dorsal and ventral margins and lateral surface; in ventral view, C-shaped, setose, lateral margin slightly concave, mesal margin concave basally and inflated subapically, posterior margin slightly truncate; second segment of inferior appendage, in lateral view, subtriangular, very short, setose, setose, dorsal margin straight, ventral margin with a very small concavity subapically, apex narrowly rounded; second segment of inferior appendage, in ventral view, digitate, slightly curved, apex rounded. Proctiger, in lateral view, narrow basally, wider apically, with a short, strong carina laterally at the base, slightly membranous basally, very long setae along basal carina, apex truncate and widely rounded posteroventrally. Phallic apparatus complex; phallotheca broadly rounded basally, phallic apodeme indiscernible; with ventral process articulating with inferior appendages, triangular; ventrolateral branches of phallotheca absent; dorsal process of phallotheca absent; posterior section of phallotheca, in lateral view, broad basally, tapering towards apex, directed posterad, apical half-covered with short, spine-like setae, lateral surface produced into a short wing, directed laterad, apex narrowly rounded; posterior section of phallotheca, in dorsal view, with a deep notch mesally; phallic spine elongate, stout, with a strong convex curvature near the base, then straight; in dorsal view, apex acute.

## Distribution

Ecuador: Zamora Chinchipe Province.

## Etymology

*Atopsyche jocotoco* is named after the Jocotoco Foundation, which owns the Tapichalaca Reserve. This conservation area protects the habitat of the endemic Jocotoco antpitta, and the amazing biodiversity in the region.

*Atopsyche papallacta* Rázuri-Gonzales, Holzenthal & Ríos-Touma, sp. nov.
LSID urn:lsid:zoobank.org:act:93B371E4-9580-4244-BD5D-B9A4CDAC2C32
Figs. 2, 8.

## Diagnosis

This is a very distinctive species among all the known Ecuadorian species and those from other regions. It shares some similarities with *A. cajas* in the short, quadrate inferior appendage, both apparently lacking the second segment. However, in other features of the inferior appendages, these two species are strikingly different. *Atopsyche papallacta* has a very prominent basodorsal process that bears strong spine-like setae, which is absent in *A. cajas*. Additionally, the second segment of the inferior appendages in the new species is secondarily divided into a pair of sclerotized, apically acute, digitate processes, while in *A. cajas*, the second segment is entire.

## Material examined

### Holotype

ECUADOR ● ♂; Napo Province, Parque Nacional Cayambe-Coca, Quebrada Piburja; 0.21242°S, 78.07785°W; 3,300 m a.s.l.; 20 Feb. 2007; B. Rios-Touma leg.; UMSP (in alcohol) [UMSP000138351].

### Paratypes

ECUADOR ● 1♂; Parque Nacional Cayambe-Coca, Río Papallacta, above Termas Papallacta; 0.35364°S, 78.15117°W; 3,386 m a.s.l.; 13 Mar. 2020; R. Holzenthal, S. Pauls, P. Frandsen and X. Amigo; SMF (pinned) ● 7♂♂; same data as the preceding; 31 Dec. 2022; B. Ríos-Touma and X. Amigo leg.; MECN (in alcohol) ● 16♂♂2♀♀; same data as the preceding; 17 Nov. 2023; B. Ríos, R. Holzenthal, P. Frandsen and X. Amigo leg.; UMSP (pinned) ● 1♀; same data as the preceding; MECN (pinned).

## Description

**Adult.** Forewing length male: 9.5–10.5 mm ($n = 7$), female: 10 mm ($n = 3$). Body brown, wings dark brown. Forewing with scattered dark brown setae and longer, erect straw-colored and black setae along major longitudinal veins; with diffuse patch of erect black setae where A2 joins A1; very small patch of cream-colored setae at apex of Cu2; small patches of cream-colored setae subapically between forks and very small patches of cream colored-setae at apices of major longitudinal veins at apex of wing from R1 through Cu2.

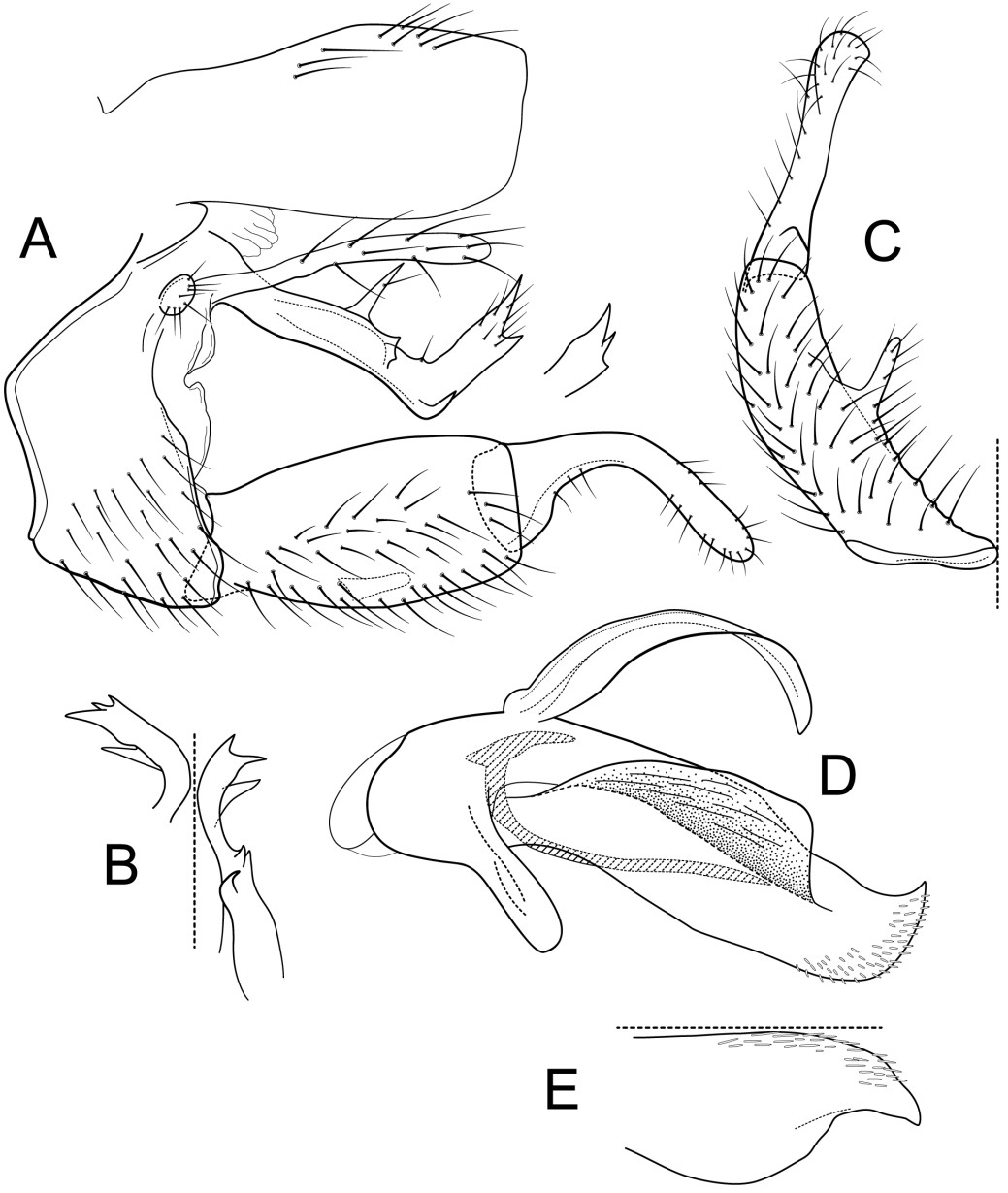

**Figure 8** *Atopsyche piburja.* Male genitalia of *Atopsyche piburja*, new species. (A) Segments IX and X, lateral. (B) Parapod apices, dorsal. (C) Inferior appendage, ventral. (D) Phallic apparatus, lateral. (E) Phallic apparatus, right side of phallotheca, ventral.

Sterna III–IV without glands; sternum V with a pair of long, membranous glands; processes on sternum VI and VII short.

**Male genitalia.** Segment IX, in lateral view, quadrangular, almost as high as long, with setae on ventral surface. Parapod, in lateral view, elongate, slightly narrower subbasally, directed posterad, apex narrowly rounded, dorsal surface with a few setae on apical third, ventral surface with a single spine-like setae; in dorsal view, elongate, almost straight, apex

rounded. Filipod digitate, longer than parapods, setose. Preanal appendage short, rounded, setose. First segment of inferior appendage, in lateral view, quadrate, dorsal and posterior margins slightly concave, ventral margin slightly convex, with a prominent basodorsal process with spine-like setae posteroventral corner not produced, with setae on ventral margin and lateral surface; in ventral view, quadrate, setose, lateral margin slightly convex, mesal margin straight, posterior margin truncate with an apical notch where the second segment of the inferior appendages are inserted; second segment of inferior appendage, in lateral view, secondarily divided into two processes: a dorsal spine-like process and a ventral L-shaped process (these processes are closer to each other in some paratypes; Fig. 8A, inset a); second segment of inferior appendage, in ventral view, bilobed, mesal lobe longer than lateral lobe, tapering to acute apex, lateral lobe with the same width throughout its length, apex truncate. Proctiger, in lateral view, narrow basally, wider apically, with a short carina laterodorsally at base, slightly membranous throughout its length, without setae, apex truncate. Phallic apparatus complex; phallotheca broadly rounded basally, phallic apodeme indiscernible; with ventral process articulating with inferior appendages, narrow, tapering towards apex; ventrolateral branches of phallotheca present, mostly linear, apex slightly capitate, approximately 0.85 times as long as posterior half of phallotheca, rounded apically; dorsal process of phallotheca absent; posterior section of phallotheca, in lateral view, broad basally, tapering towards apex, directed slightly posterad, apical third covered with small, spine-like setae dorsally and apically, apex narrowly rounded; posterior section of phallotheca, in dorsal view, with a deep notch mesally; phallic spine elongate, stout, with a slight convex curvature near the base, then slightly sinuous; in dorsal view, apex arrow-shaped.

### Distribution

Ecuador: Napo Province.

### Etymology

*Atopsyche papallacta* is named after the Papallacta town and river valley, where this species is abundant.

**_Atopsyche piburja_ Rázuri-Gonzales, Holzenthal & Ríos-Touma, sp. nov.**
LSID urn:lsid:zoobank.org:act:020715D2-D200-41AA-AF3B-A6D19F794B49
Figs. 2, 9

### Diagnosis

*Harper & Turcotte (1985)* described *A. catherinae* from the Quinuas Valley near Cuenca (Azuay province) in southern Ecuador, which is similar to a specimen we collected from Oyacachi in Cayambe-Coca National Park in northern Ecuador, some 270 km apart and in separate mountain ranges. The holotype of *A. catherinae* is a pharate male and was unavailable for study. While its illustration and description lack details, there are distinct differences between it and our specimen to recognize the latter as a new species. Both species have an elongate, curved second segment of the inferior appendage. However, in

*A. catherinae*, this segment is relatively longer than the first segment and more strongly curved than in *A. piburja* (Fig. 9). Secondly, Harper and Turcotte described the parapod of *A. catherinae* as ''antler-like'', which is also true for *A. piburja*. However, in *A. piburja* the parapod is more spinose, wider, and more angulate apically than in *A. catherinae,* and the left and right branches are asymmetrical. Harper and Turcotte did not mention any asymmetry in this appendage in their species.

## Material examined
### *Holotype*
ECUADOR ● ♂; Napo Province, Parque Nacional Cayambe-Coca, Quebrada Piburja; 0.21242°S, 78.07785°W; 3,300 m a.s.l.; 25 Jun. 2006; A. Aigaje, A. Encalada, B. Ríos-Touma leg.; UMSP (in alcohol) [UMSP000145863].

## Description
**Adult.** Forewing length male: 11 mm ($n = 1$). Body and wings light brown. Wings denuded (specimens in ethanol). Abdomen damaged, glands and processes lost.

   **Male genitalia.** Segment IX, in lateral view, quadrangular, twice as high as long, with setae on ventral surface. Parapod, in lateral view, elongate, same width throughout its length, strongly bent dorsad subapically, apex with two acute projections (these projections even vary between sides of the same individual), basally with two spine-like projections on dorsal margin and an additional spine-like projection at the bent on the ventral margin, with short setae apically; in dorsal view, elongate, length varies between sides on the same specimen, apical third with various spine-like projections apically and subapically. Filipod digitate, slightly shorter or as long as parapods, setose. Preanal appendage short, rounded, setose. First segment of inferior appendage, in lateral view, quadrate, slightly inflated posteriorly, ventral and dorsal margins almost straight, posteroventral corner not produced, with setae on ventral margin and lateral surface; in ventral view, C-shaped, setose, lateral margin convex, mesal margin very slightly concave, with a slender projection mid-length, as long as a third of the length of the first segment of the inferior appendage, posterior margin truncate; second segment of inferior appendage, in lateral view, digitate, almost as long as first segment of inferior appendage, with a few setae on ventral and dorsal margins, dorsal margin slightly convex, ventral margin concave, slightly inflated basally, apex rounded; second segment of inferior appendage, in ventral view, digitate, straight, apex rounded. Proctiger, in lateral view, almost the same width throughout, without carina, with setae on dorsal margin, apex truncate. Phallic apparatus complex; phallotheca broadly rounded basally, phallic apodeme visible, each side produced into wide flanges directed dorsolaterad; with ventral process articulating with inferior appendages, thumb-shaped; ventrolateral branches of phallotheca absent; dorsal process of phallotheca present, elongate, leaf-like, roughly half the length of the phallotheca; posterior section of phallotheca, in lateral view, broad throughout its length, directed slightly posteroventrad, apex covered with short, peg-like setae, lateral surface produced into a large flap, produced dorsolaterad, apex broad with an acute point dorsally; posterior section of phallotheca, in dorsal view, with a deep notch mesally, apex slightly directed laterad; phallic spine elongate,

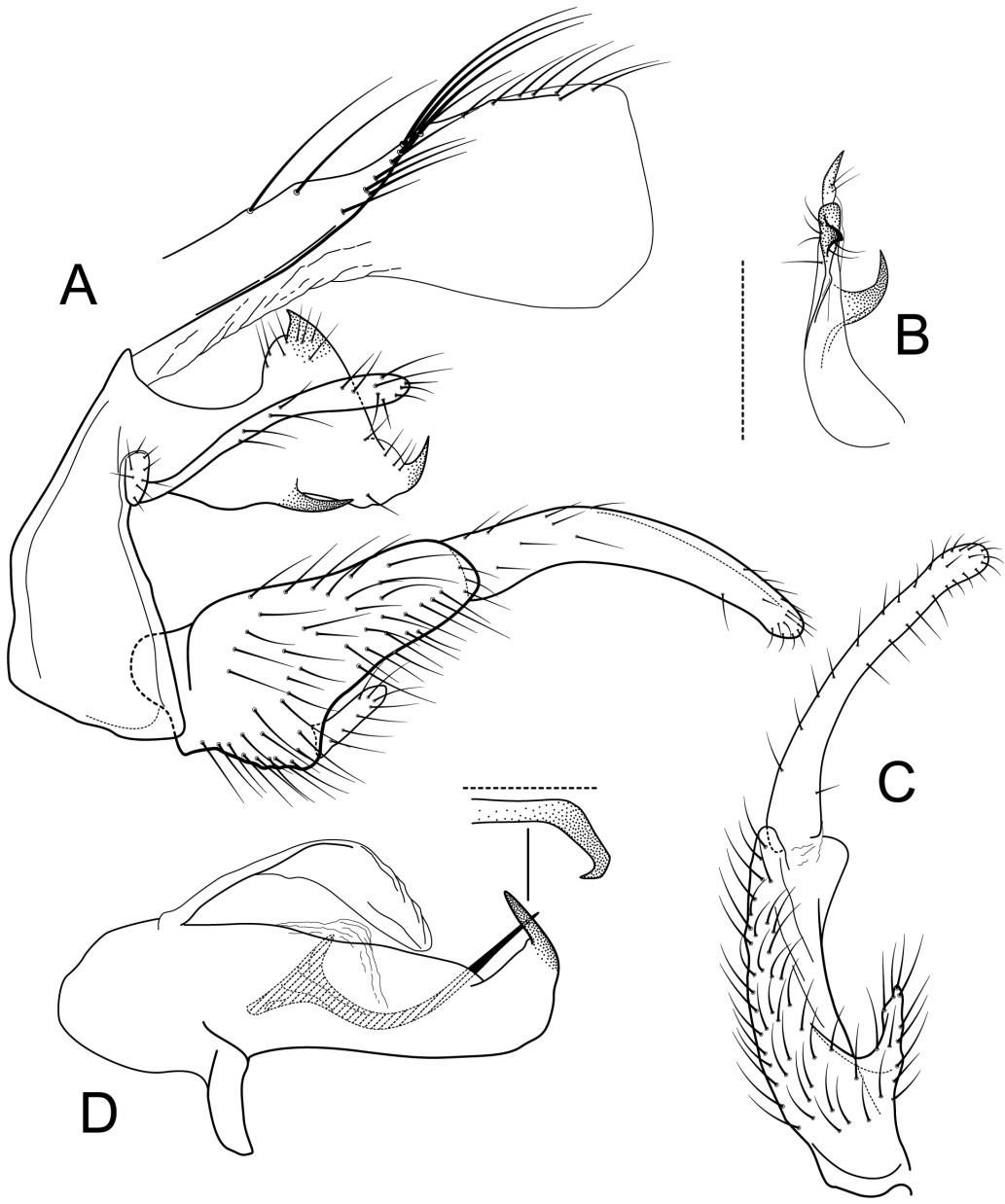

**Figure 9** ***Atopsyche tapichalaca.*** Male genitalia of *Atopsyche tapichalaca*, new species. (A) Segments IX and X, lateral. (B) Left parapod, dorsal. (C) Inferior appendage, ventral. (D) Phallic apparatus, lateral (inset, apex dorsal).

stout, with a strong convex curvature near the base, then straight; in dorsal view, apex acute.

### Distribution
Ecuador: Napo Province.

### Etymology
*Atopsyche piburja* is named after the stream where this species occurs, in the Oyacachi River valley.

**Atopsyche tapichalaca** Rázuri-Gonzales, Holzenthal & Ríos-Touma, sp. nov.
LSID urn:lsid:zoobank.org:act:2DC3C2C0-1D4A-4BD4-AD66-6CA08E43BEE0
Figs. 2, 10, 11

### Diagnosis
*Atopsyche tapichalaca* (Fig. 10) is another new species with a close relative in Ecuador, *A. bravoi*, *Gomes & Calor, 2019*. These two species are also widely separated (470 km) and occur in different mountain ranges and basins. The inferior appendages in both species have very elongate and curved first segments, but in *bravoi* the second segment is about 1.8x as long as the more ovate first segment. In *A. tapichalaca*, the second segment is only slightly longer than the first, which is more elongate than oval. There are also differences in the shape and distribution of the spine-like processes on the parapods, as seen in the illustrations (Fig. 11).

### Material examined
#### Holotype
ECUADOR ● ♂; Zamora Chinchipe Province, Tributary to Quebrada Tapichalaca on Highway E682, N of Valladolid; 4.49422°S, 79.12828°W; 2,435 m a.s.l.; 11 Feb. 2023; B. Rios, X. Amigo and J. Huisman leg.; UMSP (in alcohol) [UMSP000551055].

#### Paratypes
ECUADOR ● 2♂; same data as the holotype; MECN (in alcohol) ● 1♂; same data as the holotype; SMF (in alcohol) ● 1♂; same data as the holotype; UMSP (in alcohol).

### Additional material examined
*Atopsyche bravoi*; ECUADOR ● 1♂; Napo Province, Parque Nacional Cayambe-Coca, Río Papallacta, above Termas Papallacta; 0.35364°S, 78.15117°W; 3,387 m a.s.l.; 17 Nov. 2023; B. Ríos, R. Holzenthal, P. Frandsen, X. Amigo leg.; UMSP (in alcohol)

### Description
**Adult.** Forewing length male: 7–7.5 mm ($n = 5$). Body light brown, wings brown. Wings denuded (specimens in ethanol). Sterna III–IV without glands; sternum V with a pair of tiny protuberances; process on sternum VI long and curved, process on sternum VII shorter than the process on sternum VI and straight.

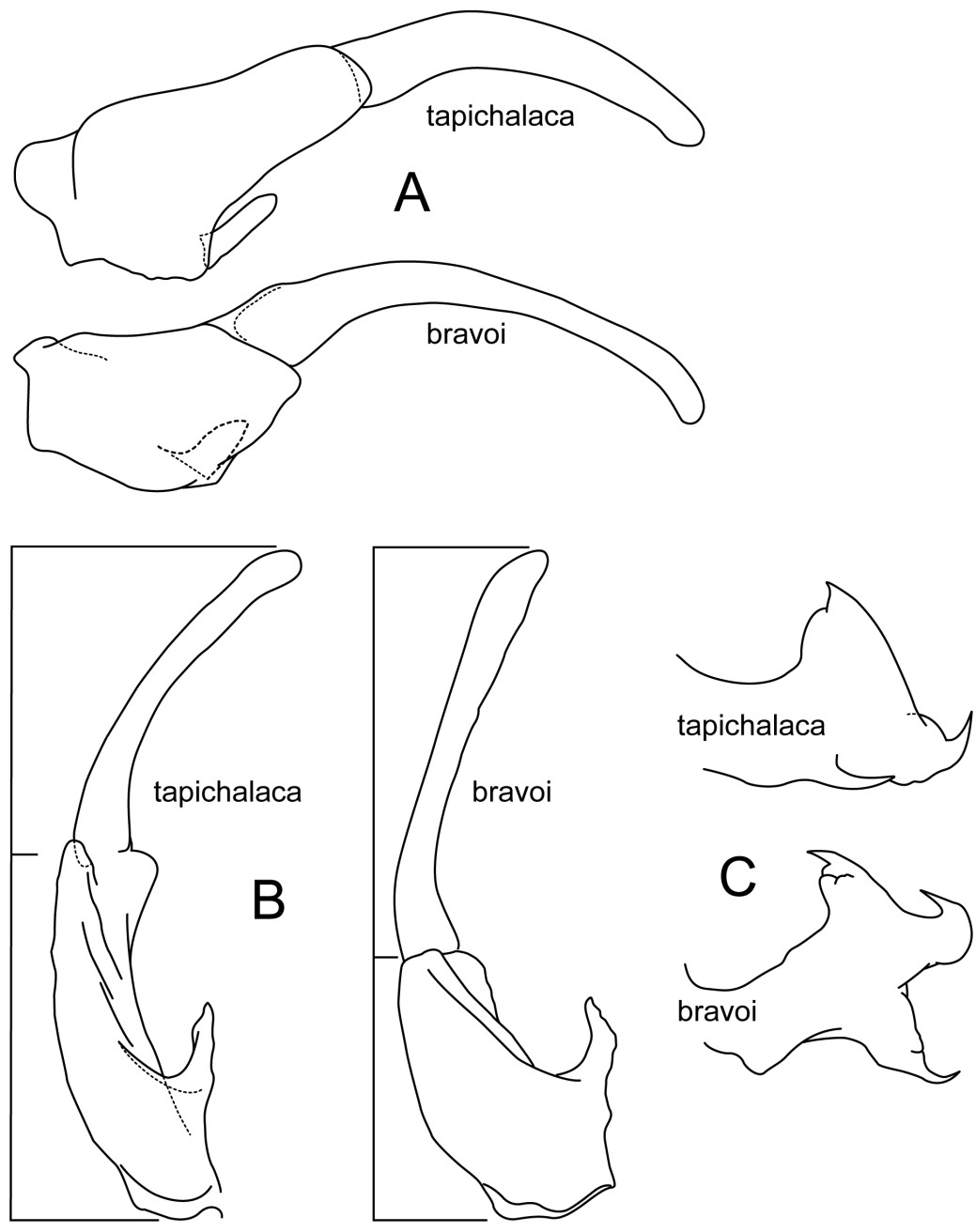

**Figure 10** *Atopsyche tapichalaca, Atopsyche bravoi.* Male genitalia, comparison. (A) Inferior appendages, lateral. (B) Same, dorsal. (C) Parapods, lateral.

**Male genitalia.** Segment IX, in lateral view, quadrangular, three times as high as long, without setae. Parapod, in lateral view, short, apical half wider than basal half and expanded into lobe with three spine-like projections (one ventrally, one dorsally, and one apically), directed posterad, apex acute, directed slightly dorsad, with short spine-like setae mostly on dorsal surface; in dorsal view, short, slightly broader basally, lateral margin slightly

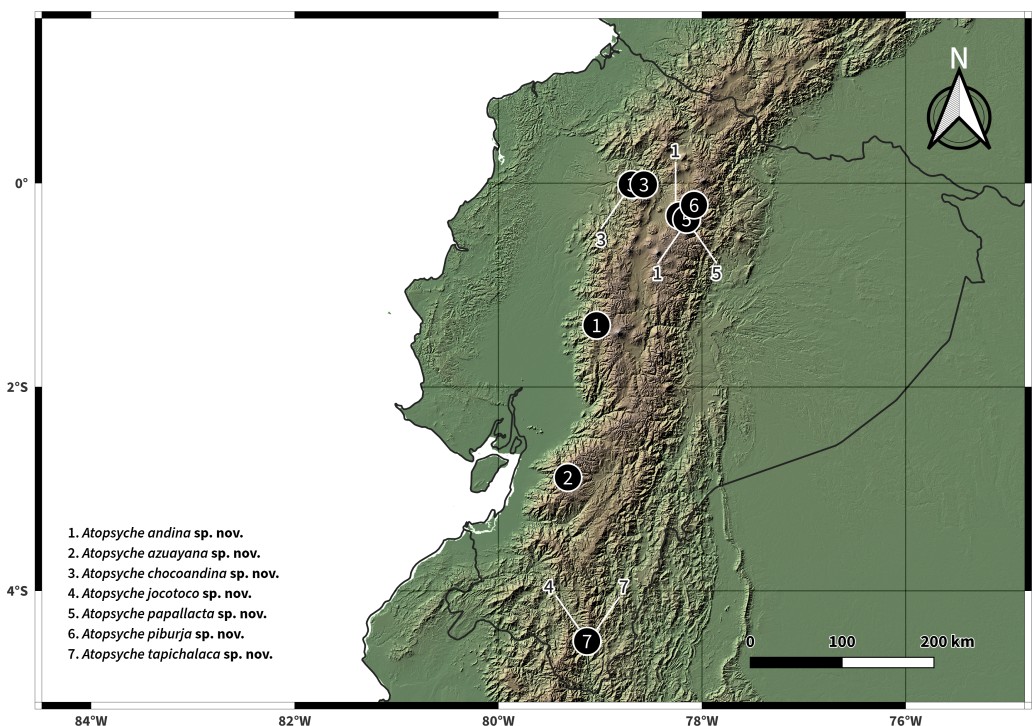

**Figure 11** Distributions of new species of *Atopsyche*.

concave and with a strong curved process, mesal margin almost straight, dorsal surface with a spine-like projection on apical third, with setae on apical third, apex acute. Filipod digitate, slightly shorter or as long as parapods, setose. Preanal appendage short, rounded, setose. First segment of inferior appendage, in lateral view, pentagonal, broader basally than apically, with setae on ventral and dorsal margins and lateral surface; in ventral view, mitten-shaped, setose, lateral margin convex, mesal margin convex with a slender projection arising basally, as long as a fourth of the length of the first segment; second segment of inferior appendage, in lateral view, digitate, slightly longer than first segment of inferior appendage, with a few setae basally and apically, dorsal margin slightly convex, ventral margin concave, apex rounded; second segment of inferior appendage, in ventral view, digitate, slightly curved, apex rounded. Proctiger, in lateral view, narrow basally, wider apically, with a long carina laterodorsally, ventral margin membranous basally, very long setae along carina and long setae on dorsal margin, apex truncate. Phallic apparatus complex; phallotheca broadly rounded basally, phallic apodeme undiscernible; with ventral process articulating with inferior appendages, thumb-shaped; ventrolateral branches of phallotheca absent; dorsal process of phallotheca present, basal half narrow, apical half broad, slightly membraneous, roughly half the length of the phallotheca; posterior section of phallotheca, in lateral view, broad basally, tapering towards apex, slightly recurved, apex slightly sclerotized and acute; posterior section of phallotheca, in dorsal view, with a deep notch mesally, apex recurved laterally; phallic spine, stout, with a strong convex curvature near the base, then straight; in dorsal view, apex acute.

### Distribution

Ecuador: Zamora Chinchipe Province.

### Etymology

*Atopsyche tapichalaca* is named after the breathtaking type locality, the Tapichalaca Reserve.

### *Diversity and distribution of Atopsyche from Ecuador*

We have documented 534 records from 116 collection events in 17 provinces, including our collection efforts and UMSP specimen holdings (Table S1). These records represent 28 species and 884 individuals, including the seven new species described in this study (Fig. 2, Table 1). The number of species in Ecuador is now 37, considering the literature records, the seven new species, and the three new records.

The most prevalent species is *A. callosa*. It is widely distributed and abundant, mainly below 1,200 m a.s.l. However, some individuals were collected at 2,460 m a.s.l. It is more common on the Pacific slope but also occurs on the Amazon slope. Another species of interest is *A. puharcocha*, exclusive to mid-elevations from 1,460 to 2,300 m a.s.l. on both slopes of the Andes. The third most abundant species is the newly described *A. andina*, only found in three localities, all above 3,300 m a.s.l. *Atopsyche copayapu* and *A. incatupac* were present in five localities, all on the Pacific slope, but they had low abundances. *Atopsyche lobosa* was also present in five localities, four on the Pacific slope and one on the Amazon slope, but in southern Ecuador, where the Andes have lower elevations. Most species generally have very low abundances, and their occurrences seem to have elevational restrictions. Eleven species were present in only one locality, and four were in two localities. Six species are only represented by one individual. Therefore, most species are rare in distribution and abundance (Table S1).

According to the CHAO 2 species estimator, the potential number of species in the country based solely on our data is 49. Therefore, we know approximately 75% of the *Atopsyche* species in the country or, stated otherwise, 12 species, likely all undescribed, are yet to be collected.

## DISCUSSION

Since 2017 (*Ríos-Touma et al., 2017*), eleven species have been added to the list of the *Atopsyche* of Ecuador, and we expect this number to increase with future efforts to study the Trichoptera fauna of the country. The endemicity pattern, higher at higher elevations, is similar to those found in other groups such as *Contulma* (*e.g.*, *Holzenthal, Ríos-Touma & Rázuri-Gonzales, 2017*). Lowland species seem to have broader distributions than high-elevation ones (Table S1). *Atopsyche callosa* is largely the most widespread in the lowlands to middle elevations. The species found above 3,000 m a.s.l. seem to be restricted to these altitudes, and middle elevations seem to harbor more species, a pattern shown in other groups of Trichoptera (*Ríos-Touma et al., 2022*). Many of these species are rare in occurrence and abundance, a pattern observed across many groups in the Neotropics (*e.g.*, *Coddington et al., 2009*; *Alroy, 2015*; *Ríos-Touma et al., 2022*).

**Table 1** Updated list of *Atopsyche* species from Ecuador.

| Species | Province | Endemic | Elevation | Source |
|---|---|---|---|---|
| *andina* sp. nov. | Bolivar, Napo, Pichincha | E | 3,300–3,850 | This study |
| *azuayana* sp. nov. | Azuay | E | 3,645 | This study |
| *banksi Ross, 1953* | Chimborazo | | 2,800 | *Sykora (1991)* |
| *bravoi Gomes & Calor, 2019* | Napo | E | 3,390 | *Gomes & Calor (2019)*; this study |
| *cajas Harper & Turcotte, 1985* | Azuay | E | 3,300–3,308 | *Harper & Turcotte (1985)*; this study |
| *callosa* (*Navás, 1924*) | Azuay, Bolivar, Carchi, Cotopaxi, El Oro, Morona Santiago, Napo, Orellana, Pastaza, Pichincha, Loja, Santo Domingo, Tungurahua, Zamora-Chinchipe | | 392–2,500 | *Sykora (1991)* and *Ríos-Touma et al. (2017)*; this study |
| *catherinae Harper & Turcotte, 1985* | Azuay | E | 3,300 | *Harper & Turcotte (1985)* |
| *chirihuana Schmid, 1989* | Pichincha | E | 229 | *Schmid (1989)* |
| *chirimachaya Harper & Turcotte, 1985* | Azuay | E | 3,300 | *Harper & Turcotte (1985)* |
| *chocoandina* sp. nov. | Pichincha | E | 2,170–2,614 | This study |
| *clarkei Flint Jr, 1963* | Morona Santiago | | 2,200 | *Sykora (1991)* |
| *copayapu Schmid, 1989* | Pichincha, Loja, Santo Domingo | E | 550–1,091 | *Sykora (1991)*; *Ríos-Touma et al. (2017)*; this study |
| *davidsoni Sykora, 1991* | Bolivar, Napo | E | 3,390–3,420 | *Sykora (1991)*: this study |
| *flinti Sykora, 1991* | Chimborazo | E | 3,500 | *Sykora (1991)* |
| *incatupac Schmid, 1989* | Azuay, Bolivar, Imbabura, Cotopaxi, El Oro, Pichincha | E | 1,150–2,500 | *Sykora (1991)*; this study |
| *janethae Harper & Turcotte, 1985* | Azuay | E | 3,300 | *Harper & Turcotte (1985)* |
| *jocotoco* sp. nov. | Zamora Chinchipe | E | 2,435 | This study |
| *kingi Ross, 1953* | Carchi | | 1,241 | New country record |
| *lobosa Ross & King, 1952* | Pichincha, Zamora Chinchipe | | 2,435–2,813 | *Ríos-Touma et al. (2017)*; this study |
| *maitacapac Schmid, 1989* | Sucumbíos, Pastaza; Napo | E | 260–1,456 | *Sykora (1991)*; this study |
| *mancocapac Schmid, 1989* | Pastaza | | 950 | *Sykora (1991)* |
| *mayucopac Schmid, 1989* | Napo | | 2,091 | New country record |
| *milenae Sykora, 1991* | Bolivar | E | 3,420 | *Sykora (1991)* |
| *neolobosa Flint Jr, 1963* | Azuay, Napo, Loja | E | 2,500–3,200 | *Flint Jr (1963)*; this study |
| *neotropicalis Schmid, 1989* | Pastaza | | 703 | New country record |
| *onorei Sykora, in Flint Jr, Holzenthal & Harris, 1999* | Loja, Zamora Chinchipe | E | 2,435–3,130 | *Sykora (1991)*; this study |

**Table 1** (*continued*)

| Species | Province | Endemic | Elevation | Source |
|---|---|---|---|---|
| *pachacutec Schmid, 1989* | Imbabura, Cotopaxi, El Oro, Pichincha | E | 900–1,770 | *Sykora (1991)*; this study |
| *papallacta* sp. nov. | Napo | E | 3,300–3,386 | This study |
| *piburja* sp. nov. | Napo | E | 3,300 | This study |
| *puharcocha Schmid, 1989* | Imbabura, Morona Santiago, Napo, Pichincha | | 1,460–2,268 | *Sykora (1991)*; this study |
| *rawlinsi Sykora, 1991* | Loja, Napo, Tungurahua, Zamora Chinchipe | E | 2,373–3,380 | *Sykora (1991)*; this study |
| *sinchicurac Schmid, 1989* | Loja, Napo, Zamora-Chinchipe | E | 1,420–2,500 | *Schmid (1989)*; this study |
| *tampurimac Schmid, 1989* | Napo, Zamora-Chinchipe | | 1,420–2,076 | *Schmid (1989)*; *Sykora (1991)*; this study |
| *tapichalaca* sp. nov. | Zamora Chinchipe | E | 2,435 | This study |
| *tlaloc Schmid, 1989* | Azuay, Zamora Chinchipe | E | 2,200–2,435 | *Schmid (1989)*; *Sykora (1991)*; this study |
| *vatucra Ross, 1953* | Morona Santiago | | 1,076 | *Ríos-Touma et al. (2017)*, this study |
| *youngi Sykora, 1991* | Azuay, Pichincha | E | 2,600–2,805 | *Sykora (1991)*; this study |

The patterns of endemism, which are more pronounced at higher elevations in the mountains, underscore the necessity for a more detailed exploration of these ecosystems. Knowledge regarding the life history of most Trichoptera species in the country remains minimal. Understanding the associations between larval stages and their habitats is crucial, especially given that Andean streams are experiencing significant degradation due to land use and climate change (*Ríos-Touma & Ramírez, 2019*). Consequently, the risk of extinction or extirpation for sensitive species such as *Atopsyche* is likely, even before we fully understand the range of species present.

### Institutional abbreviations

| | |
|---|---|
| **CMNH** | Carnegie Museum of Natural History, Pittsburgh, Pennsylvania, USA |
| **MCZ** | Museum of Comparative Zoology, Harvard University, Cambridge, Massachusetts, USA |
| **MECN** | Museo Ecuatoriano de Ciencias Naturales, Quito, Ecuador |
| **SMF** | Senckenberg Research Institute and Natural History Museum Frankfurt, Frankfurt am Main, Germany |
| **UMSP** | University of Minnesota Insect Collection, Saint Paul, Minnesota, USA |

## ACKNOWLEDGEMENTS

We would like to thank Dr. James W. Fetzner, Jr. (CMNH), Dr. Cristal Maier, Dr. Philip Perkins, and Dr. Rachel Hawkins (MCZ), and the late Dr. Oliver S. Flint, Jr. (NMNH) for loaning the type specimens of the known species. We are grateful to Los Cedros Biological Research Station (Jose DeCoux), Jose León from Fundación Jocotoco for facilitating the research in the Tapichalaca Preserve, and Richard Parson for facilitating our research

at Bellavista Cloud Forest. Adelaida Aigaje from the Oyacachi community provided fundamental help in the Piburja river samplings. Nature Experience and Xavier Amigo provided invaluable assistance in the field and logistics. Lina Pita, Steffen Pauls, Paul Frandsen, Robin Thomson, and Jolanda Huisman provided help in the field.

### Funding

This research was supported by the University of Minnesota Retirees Association Grant to Ralph Holzenthal, the University of Minnesota Insect Museum fund, and the Universidad de Las Americas project AMB.BRT.23.02 ''Mountain freshwater diversity, from taxonomy to functional genomics, and approximation from Trichoptera- Part II''; and BIOMAS Group research funds. The funders had no role in study design, data collection and analysis, decision to publish, or preparation of the manuscript.

### Grant Disclosures

The following grant information was disclosed by the authors:
University of Minnesota Retirees Association Grant to Ralph Holzenthal.
The University of Minnesota Insect Museum fund.
The Universidad de Las Americas project AMB.BRT.23.02 ''Mountain freshwater diversity, from taxonomy to functional genomics, and approximation from Trichoptera- Part II''.
BIOMAS Group research funds.

### Competing Interests

The authors declare there are no competing interests.

### Author Contributions

- Ernesto Rázuri-Gonzales performed the experiments, analyzed the data, prepared figures and/or tables, authored or reviewed drafts of the article, and approved the final draft.
- Ralph Holzenthal conceived and designed the experiments, performed the experiments, analyzed the data, prepared figures and/or tables, authored or reviewed drafts of the article, funding acquisition, and approved the final draft.
- Blanca Ríos-Touma conceived and designed the experiments, performed the experiments, analyzed the data, prepared figures and/or tables, authored or reviewed drafts of the article, funding acquisition, and approved the final draft.

### Field Study Permissions

The following information was supplied relating to field study approvals (*i.e.*, approving body and any reference numbers):

Ministerio del Ambiente y Agua del Ecuador (No. MAAE-DBI-CM-2021-0161 and 003-14-1CFAU-FLO-DNB/MA).

### Data Availability

The data with all the collections and specimens used are available in the Supplementary Files.

### New Species Registration

The following information was supplied regarding the registration of a newly described species:

Publication LSID: urn:lsid:zoobank.org:pub:53F3F8F1-8F73-4FDD-B5FC-5F929AE66411.

*Atopsyche andina* Rázuri-Gonzales, Holzenthal & Ríos-Touma, sp. nov.
LSID urn:lsid:zoobank.org:act:E2AC6B4D-F727-43E2-A458-3D1C2AB58B29.

*Atopsyche azuayana* Rázuri-Gonzales, Holzenthal & Ríos-Touma, sp. nov.
LSID urn:lsid:zoobank.org:act:44110D77-4353-4664-8C73-E7207006DFD1.

*Atopsyche chocoandina* Rázuri-Gonzales, Holzenthal & Ríos-Touma, sp. nov.
LSID urn:lsid:zoobank.org:act:F524722C-1CFD-4EFC-A37B-A277F038C3D1.

*Atopsyche tapichalaca* Rázuri-Gonzales, Holzenthal & Ríos-Touma, sp. nov.
LSID urn:lsid:zoobank.org:act:2DC3C2C0-1D4A-4BD4-AD66-6CA08E43BEE0.

*Atopsyche jocotoco* Rázuri-Gonzales, Holzenthal & Ríos-Touma sp. nov.
LSID urn:lsid:zoobank.org:act:9E54E406-AA5B-42F9-A304-D93D6AB815A0.

*Atopsyche papallacta* Rázuri-Gonzales, Holzenthal & Ríos-Touma, sp. nov.
LSID urn:lsid:zoobank.org:act:93B371E4-9580-4244-BD5D-B9A4CDAC2C32.

*Atopsyche piburja* Rázuri-Gonzales, Holzenthal & Ríos-Touma, sp. nov.
LSID urn:lsid:zoobank.org:act:020715D2-D200-41AA-AF3B-A6D19F794B49.

### Supplemental Information

Supplemental information for this article can be found online at http://dx.doi.org/10.7717/peerj.18769#supplemental-information.

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
