# Peer review of "Diversity and distribution of the caddisfly genus Atopsyche Banks, 1905 in Ecuador, with the description of seven new species (Trichoptera: Hydrobiosidae)"

_PeerJ, doi:10.7717/peerj.18769_

## Round 0.1 · original submission · Minor Revisions

Dear authors, I kindly ask you to carefully make the final corrections to the manuscript in accordance with the reviewers' comments and send the final version of the article for completion of the review.

Reviewer 1 ·

Basic reporting

Introduction: I suggest summarizing the current state of taxonomic knowledge with regards to larvae, pupae, and females of the Ecuadorian Atopsyche. For example, point out which species have been described in the larval stage and as females. Highlighting important knowledge gaps can serve to stimulate future research.

Materials and methods: Female paratype specimens are included in four of the new species descriptions but only male genitalia are described and illustrated. How were females associated with the males? Elaborate and provide rationale for including female paratypes (e.g., useful for future studies).

Species descriptions
For three of the new species (papallacta, piburja, tapichalaca) the Material examined section is placed after the Description, whereas for the other 4 new species, the Material examined section is before the Description. Revise according to journal style.

Line 41-41: “three new country records” not “three new records”

Lines 45-49: cite supporting reference(s), if possible

Lines 68-69: Reword to clarify meaning “Schmid (1989) included the 45 species he described in his revision of the Hydrobiosidae in Ross’s subgenera.” Possible rewording: “Schmid (1989) in his revision of the Hydropbiosidae placed the 45 species he described within Ross’s subgenera.”

Line 71: Meaning unclear: “…with the 154 currently recognized species assessed”. Earlier it was stated that there are 146 extant species and 1 fossil species.

Line 79: Reword: “These lights were hung in front of a white bed sheet tethered to a USB power pack,….” to, “These lights, hung in front of a white bed sheet, were tethered to a USB power pack….”

Lines 84: “authorized” not “allowed”?

Lines 87-92 (Specimen preparation and observation) – Include information on the type of microscope(s) used and magnification.

Line 216: “Fig.” not “Figs.”

Line 508: “n = 1” not “n = 4” (the only specimen listed is the holotype).

Line 563: “n = 5” not “n = 6” (based on number of specimens examined)

References: Missing Holzenthal & Cressa (2002) cited on Line 367 and missing Holzenthal et al 2017 cited on Line 639.

Table 1:
Spelling: “mayucapac” not “mayucopac”
Typo for the entry maitacapac Schmid, 1989: “Sykora 199; this study1”

Experimental design

No comment.

Validity of the findings

No comment

·

Basic reporting

The manuscript is as per the basic structure and its title is justified, It follows the basic taxonomic procedures and complete literature references from old as well as current research has been quoted. The descriptions are well supported with the relevant figures as well as raw data and tables are also provided.
Even then some minor additions are required which are already highlighted in the pdf file text version along with suggested comments for the improvement of the manuscript.

Experimental design

The authors are working on one of the most important aquatic insect orders Trichoptera and has been working especially in South America (Ecuador) since 2011. They already have added, described and reported many taxa from this region and the current manuscript is another step in this direction.

Validity of the findings

no comment

Additional comments

Already mentioned in the manuscript pdf file itself.

Reviewer 3 ·

Basic reporting

All required standards on language, literature references, background, context, and article structure are met. All figures are relevant and very well prepared.

Experimental design

The study presented is highly relevant since it helps fill the existing gaps in our knowledge of neotropical aquatic insect diversity. Undoubtedly, the investigation was performed rigorously and meets the highest standards for this kind of study. The method section is well structured and clearly describes all methods used from field sampling to specimen preparation and observation, species illustration, description, and morphological terminology used, as well as how diversity and distributional data were analyzed. The authors did comply with the rules of the ICZN, the required text is included in the Methods sections together with the LSID for the publication; for every new species the respective LSID is listed properly, and Institutions for deposit of specimens are clearly stated. Also, the necessary information on sampling permits is provided.

Validity of the findings

The authors made considerable efforts in order to provide not only new species descriptions but also an extensive and careful review of the previously known species of Atopsyche from Ecuador, their taxonomy, and distribution. Species descriptions are very detailed with all relevant information included and the diagnosis section makes clear how to differentiate each species from similar ones. Table 1 summarizes valuable information on each species known up to date from Ecuador.

Additional comments

In my opinion, the article can be accepted the way it is. Yet, my recommendation would be to add the information from which other countries, besides Ecuador, each species has been reported, and whether the larva has been described or not (and provide the respective reference). This could be included in Table 1. This additional information would increase even more the value of the paper for researchers and students working not only with caddisflies but also those who use aquatic insects in biomonitoring programs and other environmental studies in South America, and especially the Andean region.

---

## Round 0.2 · accepted · Accept

Dear authors, I congratulate you on the acceptance of this article for publication. I hope that your research will continue and that you will send it to our journal more than once.

Reviewer 1 ·

Basic reporting

The minor revisions were carried out satisfactorily.

Experimental design

no comment

Validity of the findings

no comment

·

Basic reporting

no comments

Experimental design

no comments

Validity of the findings

no comments

Additional comments

The authors have already incorporated the suggestions/additions of the reviewers.

Reviewer 3 ·

Basic reporting

no further comments

Experimental design

No further comments

Validity of the findings

no further comments

Additional comments

The authors included all the reviewers' comments and recommendations, which improved the quality of this already valuable manuscript.

This is an excellent example of a highly valuable and carefully carried out taxonomic study that constitutes an important baseline for further biological, ecological, and applied (e.g. aquatic biomonitoring) studies.